# Do Graph Neural Network States Contain Graph Properties?

**Tom Pelletreau-Duris**                                      T.A.P.PELLETREAU-DURIS@VU.NL
**Ruud van Bakel**                                                    R.VAN.BAKEL@VU.NL
**Michael Cochez**                                                    M.COCHEZ@VU.NL
*Vrije Universiteit, Amsterdam*

**Editors:** Leilani H. Gilpin, Eleonora Giunchiglia, Pascal Hitzler, and Emile van Krieken

## Abstract

Deep neural networks (DNNs) achieve state-of-the-art performance on many tasks, but this often requires increasingly larger model sizes, which in turn leads to more complex internal representations. Explainability techniques (XAI) have made remarkable progress in the interpretability of ML models. However, the non-euclidean nature of Graph Neural Networks (GNNs) makes it difficult to reuse already existing XAI methods. While other works have focused on instance-based explanation methods for GNNs, very few have investigated model-based methods and, to our knowledge, none have tried to probe the embedding of the GNNs for structural graph properties. In this paper we present a model agnostic explainability pipeline for Graph Neural Networks (GNNs) employing diagnostic classifiers. We propose to consider graph-theoretic properties as the features of choice for studying the emergence of representations in GNNs. This pipeline aims to probe and interpret the learned representations in GNNs across various architectures and datasets, refining our understanding and trust in these models.

## 1. Introduction

Graph Neural Networks (GNNs) are pivotal in harnessing graph-structured data (Kipf and Welling, 2017) for tasks ranging from social network analysis to bioinformatics. Despite their success, the black-box nature of GNNs poses significant challenges as classical XAI methods cannot be directly applied on GNNs due to the lack of a regular structure (e.g. vertices can have different degrees). In this case, explaining a prediction means identifying important parts of the relational structure, or input features of nodes. An issue is that finding the explanation is itself a combinatorial problem, making XAI methods for GNN intractable (Ying et al., 2019; Longa et al., 2023a).

The surveys by Dai et al. (2022) and Agarwal et al. (2023) highlighted the lack of comprehensive, robust and model-agnostic explainability methods. We also identified (see appendix C) few model-level explainability methods. Most works in GNN explainability present methods that identify a set of nodes and possibly their attributes as the explanation of a prediction (Vu and Thai, 2020; Zhang et al., 2021; Saha et al., 2022; Azzolin et al., 2023; Wang et al., 2023; Xuanyuan et al., 2023). Other works focus on explaining the decision-making process at a high level, often by generating graph patterns or motifs that influence the predictions (Ying et al., 2019; Yuan et al., 2020). However, oftentimes there is more information in the structure of the relationships between elements than in the juxtaposition of the elements themselves. Graph prediction ought to take into account structure that is invariant across the graph hierarchies and symmetries. Achieving this, we also get interpretability of intermediate layers, which previous methods do not provide.

One prior work identified the role probing classifiers could play (Akhondzadeh et al., 2023), as developed for Natural Language Processing (Giulianelli et al., 2018; Belinkov, 2021). That work focused on whether the hidden representation encoded the number of hydrogen atoms or the presence of aromatic rings. We aim to address the more fundamental and general question; finding out whether the hidden representation encodes information about the graph-theoretic properties. We propose graph theoretic properties as the best candidate for studying GNN inductive bias. Graph theoretic properties are natural algorithmic abstractions of graphs used in network science (Barabási et al., 2002). They would act as representational atoms (Bereska and Gavves, 2024), as features that GNNs would leverage to render graph classification problems linearly separable. The main hypothesis tested in this paper is whether GNNs leverage such algorithmic strategies to preserve, abstract, or discard structural information through the network's layers. This work bridges GNN interpretability with the broader theory of linear feature representations in neural models (Nanda et al., 2023; Bereska and Gavves, 2024). We propose a model-agnostic pipeline to interpret GNN embeddings by probing for encoded graph properties (see fig. 1) across various architectures and datasets. We investigate both local properties like betweennes centrality, as well as global properties like average path length. [1]

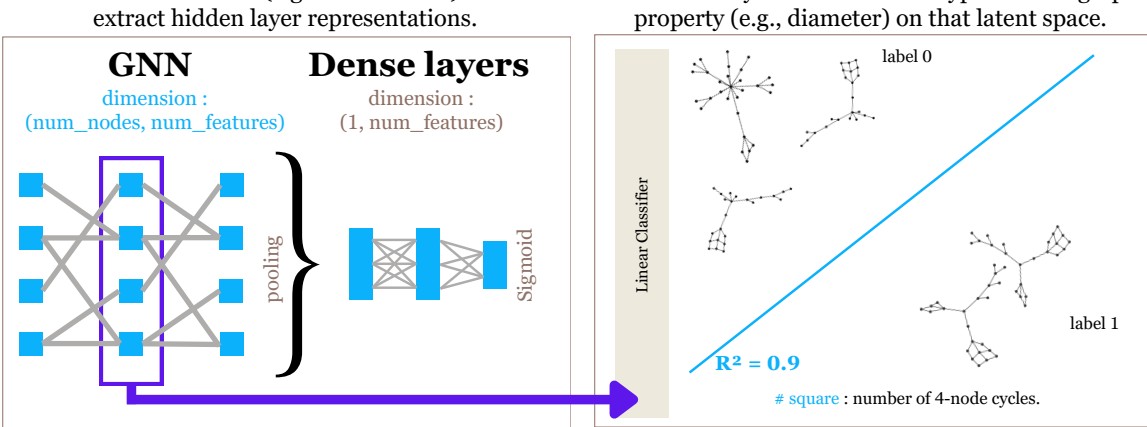

Figure 1: Illustration of the probing pipeline. Note that graphs of label 0 have only grid or house while those of label 1 have both. If a linear probe has good performance ($R^2$ score) then there exists a hyperplane in the representation space that separates the inputs based on the property.

Our core contribution is that we show that using a diagnostic classifier, as illustrated in fig. 1, we can effectively highlight graph-theoretic properties in GNN learned latent representations (fig. 7). We further explore how different regularization techniques affect the representation of graph properties (see fig. 14). We also investigate how the GNN architecture (table 6) and datasets (table 8) affect the probing.

---

1. All code is publicly available from
   https://github.com/TomPelletreauDuris/Probing-GNN-representations

## 2. Background

### 2.1. Graph Neural Networks

**Graph Convolutional Network** (GCN) ([Kipf and Welling](), 2017) are GNNs where for a single layer, the node representation is computed as: $\boldsymbol{X}' = \sigma\left(\tilde{\boldsymbol{D}}^{-1/2} \cdot \tilde{\boldsymbol{A}} \cdot \tilde{\boldsymbol{D}}^{-1/2} \cdot \boldsymbol{X} \cdot \boldsymbol{W}\right)$. We know that GNNs which rely solely on local information, like the **GCN** and its relational variant (**R-GCN**) ([Schlichtkrull et al.](), 2018), cannot compute important graph properties, such as girth and diameter or eigenvector centrality ([Garg et al.](), 2020). We are therefore also investigating more globally aware networks like **GAT** (Graph Attention Network) ([Veličković et al.](), 2018) and **GIN** (Graph Isomorphism Network) ([Xu et al.](), 2019). The models expressivity is based on the Weisfeiler-Lehman test ([Akhondzadeh et al.](), 2023). GIN aggregates node features in a way that mimics the Weisfeiler-Lehman test. By using the MLP equivalent to an Injective Update Function, GIN avoids oversimplifying the aggregation step, making it as expressive as the WL test. Thus, it is likely to excel at encoding complex graph properties and solving classification tasks. GAT should theoretically be as expressive while we expect GAT to be slightly less expressive, GCN even less.

### 2.2. Graph properties

Graph theory is a branch of mathematics that studies the properties and relationships of graphs. Graphs can be undirected or directed and analysed through both local and global properties. Local properties (like node degree or clustering coefficient) are based on the neighbors of a node. In contrast, global properties (such as diameter and characteristic path length) assess the overall graph structure. Global graph properties can be associated with higher level complex systems' characteristics like the presence of repeated motifs in the graphs or information-flow properties. See the appendix B for a list of local and global properties used in our experiments. We can distinguish different global properties, *basic* ones like the number of nodes a graph has, *clustering and centrality* ones, *graph motifs and substructures*, *spectral and small-world properties*. As an higher-order analysis, the recurrence of specific motifs within network substructures—such as triangles, cliques, or feed-forward loops can be seen as the fundamental building blocks that dictate the system's functionality and resilience. Small-worldness, as characterised by Barabási ([Albert and Barabási](), 2002), reveal how networks can maintain short path lengths despite their expansive size and sparse connectivity. GNNs synthesise local topological features into global structures, abstract these representations into higher-order graph attributes. Each layer progressively expands the receptive field, aggregating local neighborhood properties that are relevant for the classification towards high level graph properties, mirroring how hierarchical feature learning works in convolutional neural networks (CNNs) for images. Through hierarchical pooling or readout mechanisms, GNNs can aggregate node embeddings into a single, global graph-level embedding. Based on the message passing paradigm in GNNs, as layers progress, one would expect an increased abstraction in the selection of graph properties. Initially, local features like node degree should dominate, but deeper layers progressively should capture more global properties, such as connectivity patterns and centrality. Graph-theoretic properties serve a symbolic role, offering interpretable, human-defined structure,

while emergent features in GNNs reflect a connectionist process, arising through distributed representations shaped by task supervision.

## 2.3. Probing classifiers

In prior work (Hupkes et al., 2018) probing classifiers have been used for linguistic properties. The probing paradigm is a post-hoc explainability method Alain and Bengio (2018); Belinkov (2021). Probing is a concept-based approach in the larger field of mechanistic interpretability (Bereska and Gavves, 2024). Probing the learned representations for high-level concepts and patterns is a top-down approach to unravel a model's decision-making process. It's also called representation engineering (Zou et al., 2025). Here, we adapt them for graph features. Unlike unsupervised techniques such as PCA or T-SNE, which are useful to visualize input data with regard to the embedding latent space, we adopt a supervised framework to quantitatively assess how specific properties are encoded within the embedding space of GNNs. Let $g : f_l(x) \mapsto \hat{z}$ represent a probing classifier, used to map the learned intermediate representations from the original model $f$ to a specific property $\hat{z}$. The choice of a linear classifier for $g$ is motivated primarily by its simplicity. If a linear probe performs well, it suggests the existence of a hyperplane in the representation space that separates the inputs based on their properties, indicating linear separability.

The justification for simple linear probe comes from the linear representation hypothesis (Neal, 2017). The linear representation hypothesis proposes that features are directions in activation space, i.e. linear combinations of neurons (Bereska and Gavves, 2024). It's well studied in LLMs (Park et al., 2025). Rather than assuming this hypothesis holds, linear probes can be taken as diagnostic tools to investigate whether learned representations organize information in linearly accessible ways for particular concepts and datasets.

Another advantage of a simple linear probe is avoiding the risk that a more complex classifier might infer features that are not actually used by the network itself (Hupkes et al., 2018). While other non-linear probes have been explored in the literature (Belinkov, 2021), even studies showing improved performance with complex probes maintain the same logic: $\text{Perf}(g, f_1, \mathcal{D}_O, \mathcal{D}_P) > \text{Perf}(g, f_2, \mathcal{D}_O, \mathcal{D}_P)$ holds across representations $f_1(x)$ and $f_2(x)$ when evaluated by a consistent probe $g$. This consistency ensures valid comparison, underscoring that if a property can be predicted well by a simple probe, it is likely relevant to the primary classification task.

Mathematically speaking, this supervised approach allows us to define hyperplanes or higher-dimensional decision boundaries that partition the embedding space according to the chosen graph property. The $R^2$ score serves as this information-theoretic measure indicating how well the hyperplane divides the inputs in the embedding space. The $R^2$ score (see appendix A for a formal definition) measures the proportion of variance in the dependent variable that is predictable from the independent variable(s). A $R^2$ near 1 indicates that the embeddings are highly informative about $\hat{z}$, suggesting that the neural model has internalized this property in a linearly accessible manner.

By defining specific properties that could divide the embedding space and assessing how well the corresponding hyperplanes make the embedding space linearly separable, we gain quantitative insights into the abstract features aggregated within the embeddings. This method moves beyond mere hypothesis generation based on clustering patterns observed

through techniques like PCA, providing a rigorous framework for understanding how well the embedding space represents complex graph properties. It can also be thought as complementary from the T-SNE and PCA visualisation techniques. It provides a quantitative measure of the separability of the embeddings based on the property of interest. Our best empirical observation of this is the correspondence between the T-SNE visualisation of embeddings and their corresponding $R^2$ scores in figs. 3, 4 and 7.

## 3. Datasets

All three datasets have the same setup: given a set of graphs $\left\{\mathcal{G}^1, \mathcal{G}^2, \ldots, \mathcal{G}^N\right\}$, predict the corresponding binary labels $\left\{y^1, y^2, \ldots, y^N\right\}$.

**The Grid-House dataset** inspired by (Agarwal et al., 2023) is designed to evaluate the compositionality of GNNs. It features two concepts: a 3x3 grid and a house-shaped graph made of five nodes. The dataset consists of Barabási-Albert (BA) graphs (Barabási, 2009) with a normal distribution of the number of nodes. The negative class includes a BA graph connected to *either* a grid or a house, while the positive class contains a BA graph connected to *both* a grid and a house (see fig. 2). For accurate classification, models need to identify and combine simple patterns. Recognizing isolated patterns or single node features is not sufficient. In order to ensure that the classification can't be solved by simply counting the number of nodes in a graph, the average number of nodes has to be the same between classes. The number of nodes is uniformly distributed between 6 and 21 for the grid graphs, between 7 and 22 for the house graphs, and between 1 and 16 when both are present. During generation, we ensure no test set leakage by removing isomorphisms. On 2,000 graphs, we apply an 80/20 train/test split. The dataset helps investigate how GNNs combine multiple concepts and addresses the "laziness" phenomenon, where networks learn patterns characterising only one class and predict the other by default (Longa et al., 2023b).

The dataset has been structured such that an optimal, linearly separable solution requires the identification of global structural motifs in the graph. A good strategy would be counting the number of squares (i.e., four-node cycles). Important note, a random Barabási-Albert graph cannot contain any four-node cycles. Meanwhile, a grid subgraph will consistently exhibit four such cycles, while a house subgraph contains exactly one four-node cycle and one three-node cycle. Therefore, a graph that contains both a grid and a house will have a total of five four-node cycles. The presence of a three-node cycle could help the diagnostic of one type of graph in the negative class but is not necessary nor sufficient for solving the classification problem. Conversely, counting the number of four-node cycles is necessary and sufficient. In our example in fig. 3 we see that a GIN architecture organizes the input graphs in its latent space layer by layer. At layer 5 (MLP1) it even differentiates grid-only from house-only graphs in the negative class (purple).

The Grid-House dataset serves a critical purpose in our study: it offers a controlled and well-defined environment for rigorously testing our hypotheses. By construction, we are sure the classification problem can be solved in an algorithmic way. The question becomes whether GNNs inductive bias converge to this algorithmic strategy. The simplicity of these controlled constraints allow us to verify whether the GNN operates as intended in a setting where extraneous factors are minimized.

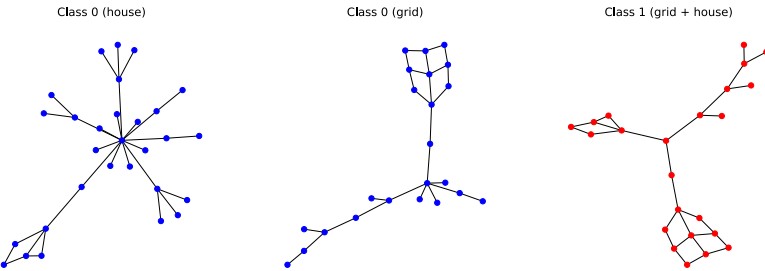

Figure 2: Illustration of the grid-house dataset. The first class (0) include graphs with either a house (square+triangle) either a 3x3 grid (4 squares). The second class include both a house and a grid.

**ClinTox Molecular** contains molecular graphs representing compounds with binary labels indicating whether they are toxic or non-toxic. The dataset consists of 1,491 drug compounds with known chemical structures. Each molecule is represented as a graph where nodes correspond to atoms and edges to bonds, with node features representing atom types and edge features representing bond types. The task is to predict toxicity.

## 4. Methodology

We propose a pipeline to confirm that graph-theoretic properties are a good candidate for studying GNN inductive bias. For each of the three datasets, we compare different GNN architectures (GCN, GIN, or GAT). The first layers are followed by a pooling operation (mean- (Kipf and Welling, 2017), sum-(Xu et al., 2019), or max-pooling (Hamilton et al., 2017)), and then a number of dense layers. For the **Grid-House dataset** the full hyperparameter settings after optimization can be found in table 4. Because we can only probe on one model weights, we ran each model 20 times and took the one with the best accuracy. There is a correlation of 0.992 between the accuracy and the maximum probing score fig. 6.

We also compare different regularization methods and their effect on GNN representations. It is known that explicit $L_2$ **regularization** encourages the network to keep the weights small. We expect that this will make the embeddings less sensitive to fluctuations in the input data which would translate in later layers being more selective to graph properties, leading to more specific receptive fields. **Dropout** randomly disables a fraction of the neurons during each training iteration which forces the network to learn redundant representations, as any neuron could be dropped out. These redundant representations might make it more difficult to linearly separate the graph embeddings. We expect later layers to distinguish less between graph properties. In other words, we expect more polysemanticity (Bereska and Gavves, 2024). We plot only post-pooling layers for the sake of clarity.

As a second set of experiments, we use our build pipeline to probe GNNs on real-life datasets, wondering if what we find aligns with domain knowledge in chemistry. For the **ClinTox Molecular** dataset, we ranged the number of layers from 4 to 6 and hidden

dimensions from 64 to 256. The final model architectures were selected based on optimal performance on this dataset. We also did preliminary work on Knowledge Graphs and fMRI connectomes, which is available in the appendix.

Before probing, we first sort the embeddings in descending order based on their norms before concatenating fig. 3. This ordering depends only on the inherent properties of the embeddings themselves, not on their original ordering in the graph. As such, it inherently respects permutation invariance because reordering the nodes does not affect their norms or the resulting sorted order. Conveniently, sorting in this way ensures that any padding zeros align at the end of the sequence, enabling learnable representations for graphs with varying node counts.

## 5. Results

### 5.1. Grid-House dataset

All tested models achieve a test accuracy of over 0.9, with the GIN without regularization or dropout reaching the highest score of 1.0 (see table 5 in the appendix for more details). The probing results in fig. 3 demonstrate that the *number of squares* consistently yields the highest $R^2$ scores. This focus on *#squares*, effectively partitions the graphs into two classes: those with #squares $< 5$ (indicating either the grid or house alone) and those with #squares $= 5$ (indicating the presence of both substructures). We see the same results for the GAT model (fig. 4), especially for the final layers and for other model variants (see the appendix). This confirms our initial hypothesis where the number of square is the property of interest to perform this classification. Interestingly, the *MLP1* layer of the GIN model can effectively separate the graph embeddings using the *#triangles* feature.

### 5.2. ClinTox Molecular

As expected, the GIN model outperform the other models with a test accuracy of 0.93 (table 8 in the appendix). We find that the highest performance is consistently achieved by probes for the features of *average degree*, the *spectral radius*, the *algebraic connectivity* and the *density*, in that order (see table 1).

These findings validate our methodology on already known domain knowledge. Indeed, the average degree of atoms in a molecule provides a straightforward interpretation, as atoms with higher valencies are generally less stable and less biologically compatible. For instance, hydrogen with a valency of 1 and oxygen with a valency of 2 are more compatible with carbon-based molecules, whereas sulfur, with a valency of 6, is less favorable for biological systems (Komarnisky et al., 2003). Therefore, the average degree serves as a useful indicator of molecular toxicity. Additionally, the spectral radius, often associated with molecular stability and reactivity, is another valuable graph property. Molecules with a lower spectral radius tend to be more stable, while those with a higher spectral radius may exhibit localized electron densities, increasing their reactivity. Domain knowledge on molecular toxicity align with the GIN strategy.

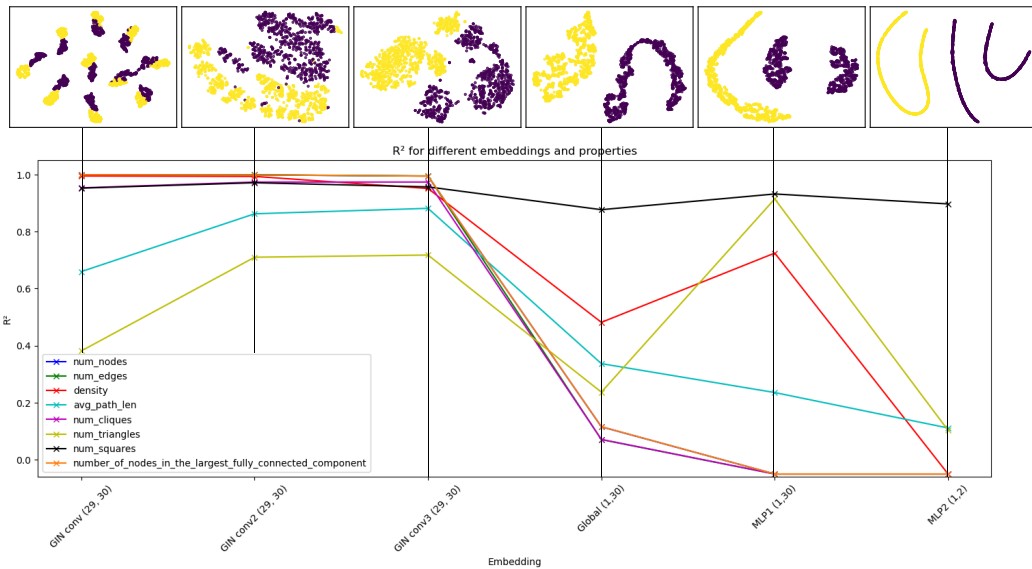

Figure 3: GIN probing results $R^2$ across different layers aligned with the T-SNE visualisations of the embedding (Grid House)

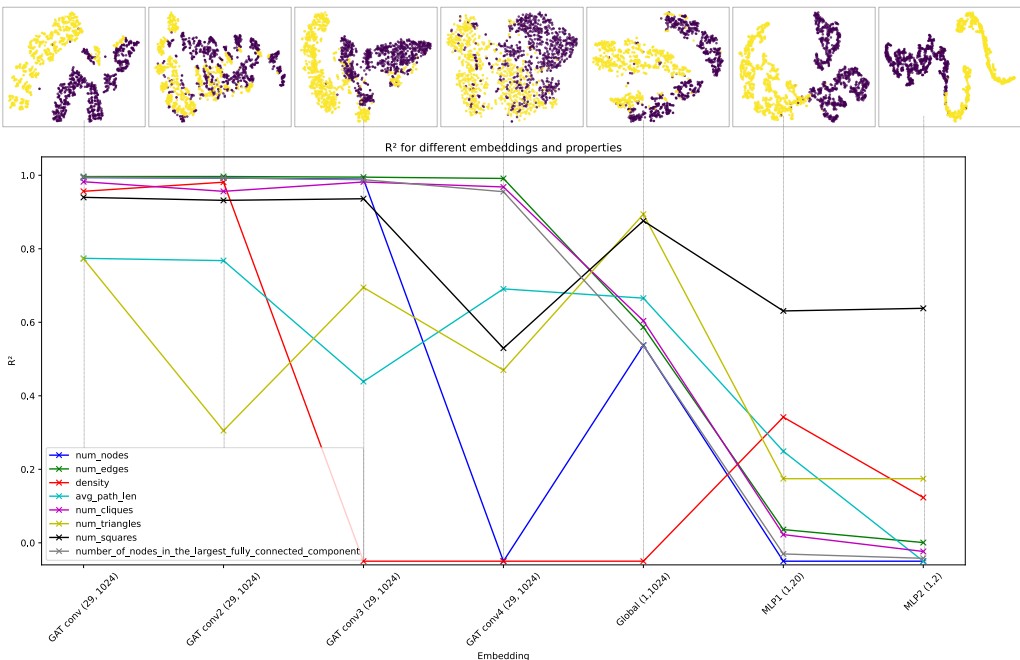

Figure 4: T-SNE visualization across different layers of our GAT architecture aligned with the probing $R^2$ scores plots (Grid House)

## 5.3. Pooling

While have deferred the results for regularization and dropout to the appendix, we did notice that regularization tends to help separate out the #squares better, while dropout has the opposite effect. This is in line with our expectations.

## 6. Discussion

**Expectations:** With *Grid-House* we hypothesized that the GNNs would benefit from leveraging the *# of square* to render the problem linearly separable. Based on their mathematical restrictions, we hypothesized that the GIN would perform better than the GAT and the GCN. Regularization methods should either refine the representations, as seen with $L_2$ regularization, or distribute them more broadly, as achieved with dropout. Based on the message-passing paradigm, we anticipated a clear absence of the *# of square* in the first layer. Additionally, we expected that the mean-pooling and norm-sorting methods would not significantly alter how representations are probed, except for basic properties like the number of nodes, which are easily interpretable from the tensor of node vectors but not from an aggregated representation. For the *ClinTox Molecular* dataset, based on the literature (Chen et al., 2021; Jiang et al., 2021; Kengkanna and Ohue, 2024) some properties have been found to be link with toxicity such as the node degree (i.e. the valency), subgraph patterns (functional groups, chemical fragments), and the overall graph connectivity.

    **Findings:** We first demonstrate the feasibility of our probing method through the *Grid-House* dataset. In line with our expectations, The *number of squares* metric dominated across all layers and models, with GIN showing enhanced expressivity. Secondary properties like *avg_path_lenght* figs. 7 and 8 showed early significance but gradually diminished through the layers, demonstrating how GNNs act as low-pass filters on graph signals. The receptive field selects the most discriminative properties. These findings align with the labels; density or average path length increases when both a grid and a house are added to a graph.

    The inductive bias of Graph Neural Networks (GNNs), when viewed through the lens of the linear representation hypothesis, reflects a tendency to organize internal representations such that task-relevant features become increasingly linearly decodable as a function of the supervision signal. In this framework, the supervision signal guides the layer-wise development of representations, selectively reinforcing directions in activation space that align with informative graph-theoretic properties (e.g., centrality, motif counts, degree distributions).

Table 1: Linear Probing $R^2$ Performance Across GIN Layers for Selected Graph Properties (ClinTox Dataset). Best Scores in Bold; Non-convergence indicated by —

| GIN Layer | Avg. degree | Sp. radius | Alg. co. | Density | Avg. btw. cent. | Graph energy |
|-----------|-------------|------------|----------|---------|-----------------|--------------|
| x_global | **0.81** | 0.74 | 0.67 | 0.58 | 0.48 | 0.44 |
| x6 (MLP) | **0.80** | 0.74 | 0.66 | 0.58 | 0.42 | 0.44 |
| x7 (MLP) | **0.75** | 0.71 | 0.56 | 0.50 | 0.47 | 0.46 |
| x8 (MLP) | — | **0.07** | 0.02 | 0.00 | 0.06 | 0.05 |

Using the *ClinTox Molecular* dataset to assess molecular toxicity, we explored how key graph properties, such as the *average degree* and *spectral radius*, are utilized by our GIN architecture. The average degree, closely linked to atomic valency, reflects a molecule's potential for interactions. The *spectral radius* offers a complementary hypothesis, suggesting that the overall structural stability of a molecule, independent of specific atomic features, may also be a key factor in toxicity prediction.

## 7. Future work

Our methodology has several limitations. While we addressed dataset issues such as leakage and isomorphic graphs, a key challenge remains the lack of guarantees that GNNs find globally optimal solutions, despite their theoretical capacity as universal function approximators (Hornik et al., 1989). As we observed, early layers in the network contain predictive information for some global graph features. It is however unclear why this is the case. They might be somehow predictive for the features we are looking for, perhaps in the specific combination with the specific distribution of graphs we used int he experiments. Investigating additional graph properties like girth or complex motifs could be beneficial. Preliminary work on alternative architectures (e.g., GATv2, GraphSAGE, ChebNet, Set2Set, HO-Conv, DiffPool) has begun but is not yet complete. An extensive exploration of 1-WL, 2-WL and 3-WL GNN equivalent could bolster the paper's contributions by showing clear restrictions and capabilities of these models. As a future encouraging work, studying the supervision signal comparing unsupervised, self-supervised and supervised models would be very insightful. Interesting application research could be done on Knowledge Graphs. Another path could be fMRI connectomes, which are also graphs. With fMRI data, some functional connectivity patterns could be associated with cognitive states under the lens of GNNs probing.

## 8. Conclusion

We demonstrate the relevance of using a probing classifier as a model-agnostic explainability method for graph neural networks. We manifest both the expressivity of different GNN architectures and their capacity to solve a graph classification problem through effective feature extraction. They render it linearly separable in the space of their embeddings through the computation of graph properties. We validate domain knowledge with the Clintox Molecular dataset. There is a manifest emergence of molecular qualities like toxicity with regard to their structural properties like *node degree* (atom valency) and *spectral radius* (the molecule's stability). To explain a macro attribute, there are instances where structural properties may offer more insight than the mere aggregation of element properties. This provide encouraging results for studying the possibility of formulating hypotheses on the emergent dependence of complex systems attributes to basic and more higher level structural properties. We could explore how the macroscopic behavior of complex systems emerges from the intricate interplay of their microscopic components, raising hypothesis on graph similarly than what is now done on language with LLMs or on vision with computer vision.

## Acknowledgment

This work is based on the MSc. AI thesis by Tom Pelletreau-Duris, a large part of the results were also presented in that work.

Michael Cochez is partially funded by the Elsevier Discovery Lab, partially funded by the Graph-Massivizer project, funded by the Horizon Europe programme of the European Union (grant 101093202), and supported by a gift from Accenture LLP. His work on this publication is in part based upon work from COST Action CA23147 GOBLIN - Global Network on Large-Scale, Cross-domain and Multilingual Open Knowledge Graphs, supported by COST (European Cooperation in Science and Technology, https://www.cost.eu).

We would also want to thank the anonymous reviewers for their comments and suggestions that helped us improve the manuscript.

Author Contributions

Pelletreau-Duris Tom: Conceptualization (lead); methodology (lead); software (lead); formal analysis (lead); writing – original draft (lead); writing – review and editing (equal). Ruud van Bakel: writing – review and editing (equal); Michael Cochez: writing – review and editing (equal); Supervision; Funding acquisition

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

## Appendix A. $R^2$ Score

We are using $R^2$ as the main metrics. The $R^2$ score (coefficient of determination) measures the proportion of variance in the dependent variable that is predictable from the independent variable(s). For a probing classifier, it would indicate how well the probe's predictions match the actual properties being probed. More formally, $R^2$ is defined as:

$$R^2 = 1 - \frac{\sum_i \left(z^{(i)} - \hat{z}^{(i)}\right)^2}{\sum_i \left(z^{(i)} - \bar{z}\right)^2}$$

Where: $z^{(i)}$ is the ground truth value of the property $\hat{z}$ for the $i$-th data point in the probing dataset $\mathcal{D}_P$. $\hat{z}^{(i)}$ is the predicted value of $\hat{z}$ produced by the probing classifier $g$. $\bar{z}$ is the mean of the ground truth values $z^{(i)}$ over the dataset. The numerator represents the residual sum of squares (how far off the predictions are), and the denominator represents the total sum of squares (the variance in the ground truth values).

An $R^2$ value ranges from 0 to 1, where: $R^2 = 1$ means the probing classifier perfectly explains the variance in the target property (i.e., the learned representations fully capture the property). $R^2 = 0$ means the probing classifier does no better than predicting the mean $\bar{z}$, implying the representations do not capture any useful information about the property. Good R2 score should indicate how the model achieves its behavior on the original task Hupkes et al. (2018).

A good $R^2$ score gives a sense of how well the features at each layer can be separated linearly to predict the target labels. The second reason is that a more complex probe "bears the risk that the classifier infers features that are not actually used by the network" (Hupkes et al., 2018). Of course, other non linear probes have been explored in the literature (Belinkov, 2021). If a few studies observed better performance with more complex probes, the logic remained the same $\text{Perf}(g, f_1, \mathcal{D}_O, mathcalD_P) > \text{Perf}(g, f_2, \mathcal{D}_O, \mathcal{D}_P)$, of two representations $f_1(x)$ and $f_2(x)$, holds across different probes $g$. The important criteria is to compare the results obtained by the same measurement system. In general, if we can predict one property on one embedding for a given classification problem, then it means this properly is useful for the problem resolution.

From an information-theoretic perspective, training the probing classifier $g$ can be viewed as estimating the mutual information between the learned representations $f_l(x)$ and the property $z$. This mutual information is denoted as $\text{I}(\mathbf{z}; \mathbf{h})$, where $\mathbf{z}$ refers to the property and $\mathbf{h}$ represents the intermediate representations (Belinkov, 2021).

## Appendix B. Local and global graph properties

| | Property | Visual Pattern & Definition | Computational Criteria |
|---|---|---|---|
| **Local** | Degree | How many links a node has which is the simplest form of centrality | Count edges per node |
| | Local clustering Coefficient | Are the neighbours of a node also connected together ? | Count triangles of neighbours / total possible triangles of neighbours |
| | Betweenness Centrality | How much of a bridge between clusters is a node. Removing that node would break many shortest paths. Importance in information flow | Number of shortest paths through node |
| | Closeness Centrality | Being in the middle of the network, the barycenter of the graph. | The average length of the geodesic distances to all the other nodes (inverse sum of shortest paths) |
| | Eigenvector Centrality | Being connected to well connected nodes without necessarily having a large number of neighbours itself; influence based on connections | Recursive definition based on neighbours |
| | PageRank | Nodes with important connections; web-inspired importance | Similar to Eigenvector but with random walk and teleportation |

Table 2: Local Network Properties with definition and computational criteria

| | Property | Visual Pattern & Definition | Computational Criteria |
|---|---|---|---|
| **Global** | Number of Nodes | Graph size; total nodes in the network | Count vertices |
| | Number of Edges | Graph density; total connections in the network | Count connections |
| | Density | Overall graph connectivity; how densely connected | Ratio of actual to possible edges |
| | Average Path Length | On average, how close are nodes to each other? Typical distance between node pairs | Average number of steps along the shortest paths for all possible pairs of nodes |
| | Diameter | Graph span; longest of all shortest paths | Maximum shortest path |
| | Radius | Graph core; minimum distance from central to farthest node | Minimum eccentricity |
| | Transitivity | Triangle density; probability of connected node triplets | Ratio of triangles to triads |
| | Assortativity | Node degree correlations; tendency of similar nodes to connect | Pearson correlation of degrees |
| | Number of Cliques | Dense subgraphs; count of maximal fully connected subgraphs | Number of maximal complete subgraphs |
| | Number of Triangles | Local density; fully connected 3-node subgraphs | Count 3-node cliques |
| | Number of Squares | 4-node patterns; cycles in the graph | Count 4-node cycles |
| | Largest Component Size | Main connected structure; size of biggest connected part | Largest set of connected nodes |
| | Average Degree | Overall connectivity; average connections per node | Mean of all node degrees |
| | Spectral Radius | Dominant graph structure; overall connectivity measure | Largest eigenvalue of adjacency matrix |
| | Algebraic Connectivity | Graph cohesion; measure of how well-connected the graph is | Second smallest eigenvalue of Laplacian |
| | Graph Energy | The eigenvalues capture deviations from regularity in the network. Complete graphs or highly connected networks tend to have higher energies due to the larger magnitude of their eigenvalues. Graph energy can help assess robustness, synchronizability. | Sum of absolute Laplacian eigenvalues |
| | Small World Coefficient | Balance of clustering and paths; small-world characteristics | Comparison to random graph |
| | Small World Index | Refined small-world measure; comparison to random and lattice graphs | Comparison to random and lattice graphs |
| | Betweenness Centralization | Central node dominance; degree of central bridging node | Variation in betweenness centrality across nodes |
| | PageRank Centralization | Influence concentration; degree of dominant influential nodes | Variation in PageRank values across nodes |

Table 3: Global Network Properties with definition and computational criteria

We are using the Small-World Index, $SWI = \left(\frac{L-L_l}{L_r-L_l}\right) \times \left(\frac{C-C_r}{C_l-C_r}\right)$ in our experiment because it provides a more balanced and robust measure of small-world properties. Unlike the Small-World Quotient: $Q = \frac{C/C_r}{L/L_r}$, which can be sensitive to network size and degree, $SWI$ normalizes both the clustering coefficient and average path length with respect to both random and lattice reference graphs. This dual normalization approach ensures that $SWI$ is less prone to false positives or negatives, making it a more reliable metric for our analysis (Neal, 2017).

## Appendix C. Literature review on related Work

Existing post-hoc GNN explanations methods can be classified into two main categories: *instance-level* and *model-level* methods (Barredo Arrieta et al., 2020). See (Agarwal et al., 2023; Dai et al., 2022) for nice reviews on the subject. In the realm of instance based methods, *gradient-based* methods use the gradients of the output with respect to the input or intermediate features to measure the importance of each component of the graph. *Decomposition-based* methods try to decompose the input graph into smaller subgraphs or paths that can account for the output. *Surrogate-based* methods use a simpler, more interpretable model to approximate the behavior of the original GNN and provide explanations based on the surrogate model. And finally *Perturbation-based* methods which perturb the input graph by removing or adding nodes, edges, or features, and observe the changes in the output to identify the influential components. The most mainstream technique, GNNExplainer (Ying et al., 2019) achieves explanation by removing redundant edges from an input graph instance, maximizing the mutual information between the distribution of subgraphs and the GNN's prediction. It is able to provide an explanation both in terms of a subgraph of the input instance to explain, and a feature mask indicating the subset of input node features which is most responsible for the GNN's prediction.

For *model-based* techniques, few methods come to mind (Saha et al., 2022; Azzolin et al., 2023; Vu and Thai, 2020; Wang et al., 2023; Xuanyuan et al., 2023; Yuan et al., 2020; Zhang et al., 2021). The most mainstream method seems to be XGNN (Yuan et al., 2020). The authors of XGNN investigate the possible input characteristics used by a GNN for graph classification. But they formulate the problem as a reinforcement learning problem and generate graph patterns iteratively. Such an iterative approach is often intractable for large graphs. Moreover, it does not allow for both node classification and graph classification explanations, nor does it allow for an investigation of the learning process through the different layers of the GNN. In general, none of the techniques allow for an interpretation of the hidden representations states with graph properties.

## Appendix D. Grid House artificial dataset

### D.1. Grid House figures

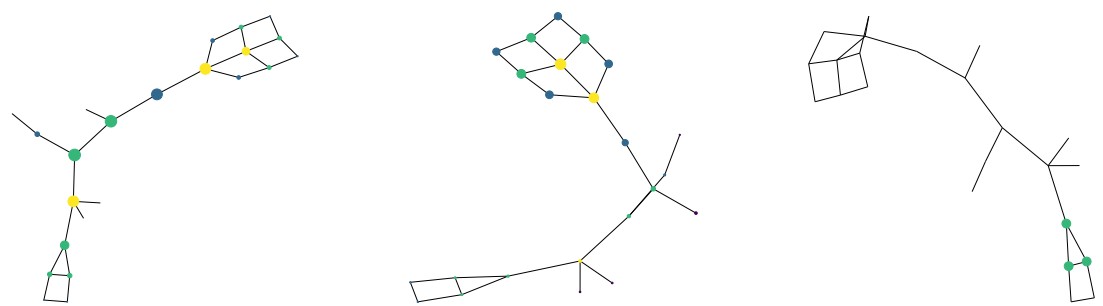

Figure 5: Comparison of different centrality measures for the first graph in our Grid House dataset: (a) betweenness centrality, (b) eigenvector (PageRank) centrality, and (c) local clustering coefficients.

Experiments [2]

## D.2. Grid House models

Table 4: Range of Hyper-parameters and Final Specification for the Grid-House Dataset

| Hyper-parameter | Range Examined | Final Specification |
|---|---|---|
| Graph Encoder | | |
| #GNN Layers | $\{[2, 3, 4, 5]\}$ | 4 (GCN), 2 (GIN), 3 (GAT) |
| #MLP Layers | $\{[2, 3, 4]\}$ | 3 (GCN), 2 (GIN), 2 (GAT) |
| Hidden Dimensions | $\{[10, 15, 30, 45, 60, 64, 128, 256]\}$ | 60 (GCN), 30 (GIN), 128 (GAT) |
| Attention Heads (GAT) | $\{[4, 8, 16]\}$ | 8 heads, 32 dimensions per head |
| Learning Rate | $\{[1e-2, 1e-3, 1e-4]\}$ | $1e-3$ |
| Batch Size | $\{[32, 64, 128, 256]\}$ | 64 |
| Weight Decay (when added) | $\{[1e-4, 1e-2]\}$ | $1e-4$ (GCN), $1e-2$ (GIN) |
| Batch Normalization | $\{with, without\}$ | $without$ |
| Dropout (when added) | $\{[0.15, 0.5]\}$ | 0.2 |
| Pooling Method | $\{mean, sum, max\}$ | $max$ (GCN), $mean$ (GIN), $max$ (GAT) |

Table 5: Performance of Different Models with Regularization on the Artificial Dataset (80%-20% Random Split). The highest performance is highlighted with boldface. All performances are reported under their best settings and rounded to 2 decimal places.

| Method | Test Accuracy |
|---|---|
| GCN (control) | 0.90 |
| GCN ($L_2$) | 0.97 |
| GCN (dropout) | 0.93 |
| GIN (control) | **1.00** |
| GIN ($L_2$) | 0.99 |
| GIN (dropout) | 1.00 |
| GAT | 0.97 |

As expected the RGCN outperform the GCN on this node classification task.

---

2. All the experiments have been done on the dutch supercomputer SNELLIUS (SURF) using GPU and CPU.

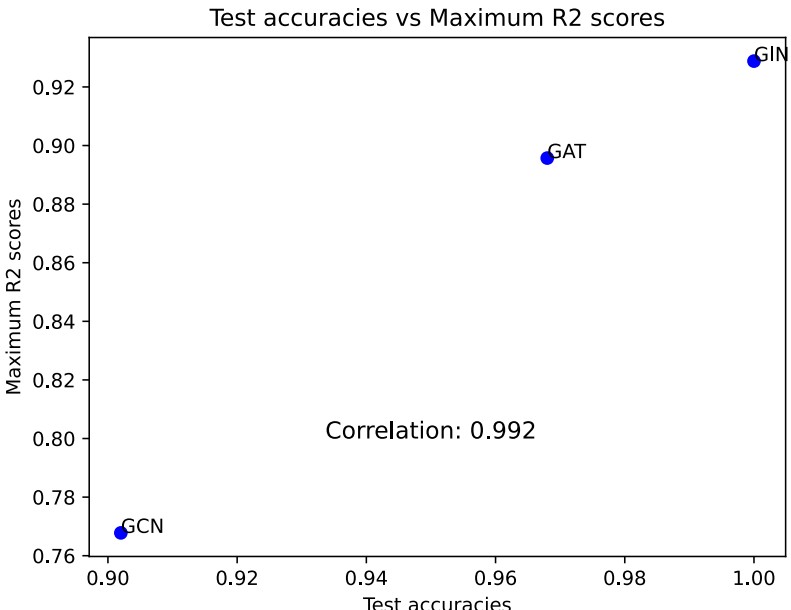

Figure 6: Plot of the correlation between the different model test accuracies and their maximum R2 score (Grid House)

## D.3. Grid House RESULTS

### D.3.1. GRAPH PROPERTIES PROBING RESULTS

Table 6: Linear Probing $R^2$ Performance Across models for Selected Graph Properties (GridHouse Dataset). Best Scores in Bold; Non-convergence indicated by —

| Model | #nodes | #edges | density | avg path len | #cliques | #triangles | #squares | #Largest Component |
|---|---|---|---|---|---|---|---|---|
| **GCN (control)** | | | | | | | | |
| x_global | 0.36 | — | 0.66 | 0.33 | 0.02 | 0.31 | **0.77** | 0.36 |
| x5 | 0.33 | 0.22 | 0.64 | 0.29 | 0.27 | 0.39 | **0.77** | 0.33 |
| x6 | 0.19 | 0.08 | 0.56 | — | 0.07 | 0.06 | **0.74** | 0.19 |
| x7 | — | — | 0.45 | 0.13 | — | 0.03 | **0.72** | — |
| **GCN ($L_2$)** | | | | | | | | |
| x_global | 0.36 | 0.09 | 0.67 | 0.35 | 0.20 | 0.68 | **0.86** | 0.36 |
| x5 | 0.31 | 0.32 | 0.66 | 0.32 | 0.32 | 0.80 | **0.86** | 0.31 |
| x6 | 0.04 | — | 0.41 | 0.15 | 0.03 | 0.23 | **0.83** | 0.04 |
| x7 | — | — | 0.29 | 0.27 | — | 0.09 | **0.81** | — |
| **GCN (dropout)** | | | | | | | | |
| x_global | 0.21 | 0.07 | 0.67 | 0.33 | 0.07 | 0.63 | **0.72** | 0.22 |
| x5 | — | — | 0.59 | 0.26 | — | 0.66 | **0.74** | — |
| x6 | — | — | 0.42 | 0.21 | — | 0.49 | **0.65** | — |
| x7 | — | — | 0.35 | 0.10 | — | 0.26 | **0.51** | — |
| **GIN (control)** | | | | | | | | |
| x_global | 0.12 | 0.07 | 0.50 | 0.32 | 0.07 | 0.22 | **0.87** | 0.12 |
| x5 | — | — | 0.72 | 0.30 | — | 0.89 | **0.93** | — |
| x6 | — | — | — | 0.02 | — | 0.11 | **0.88** | — |
| **GIN ($L_2$)** | | | | | | | | |
| x_global | — | — | 0.49 | 0.30 | — | 0.18 | **0.85** | — |
| x5 | — | — | 0.51 | 0.15 | — | 0.52 | **0.89** | — |
| x6 | — | — | 0.40 | 0.12 | — | 0.10 | **0.80** | — |
| **GIN (dropout)** | | | | | | | | |
| x_global | — | — | 0.53 | 0.36 | — | 0.25 | **0.87** | — |
| x5 | — | — | 0.71 | 0.33 | — | 0.85 | **0.93** | — |
| x6 | — | — | — | 0.21 | — | 0.34 | **0.91** | — |
| **GAT** | | | | | | | | |
| x_global | 0.54 | 0.59 | — | 0.49 | 0.61 | **0.89** | 0.87 | 0.54 |
| x5 | — | — | 0.33 | 0.27 | — | 0.17 | **0.64** | — |
| x6 | — | — | 0.25 | 0.17 | — | 0.17 | **0.63** | — |

### D.3.2. Graph properties probing plots

**GCN**

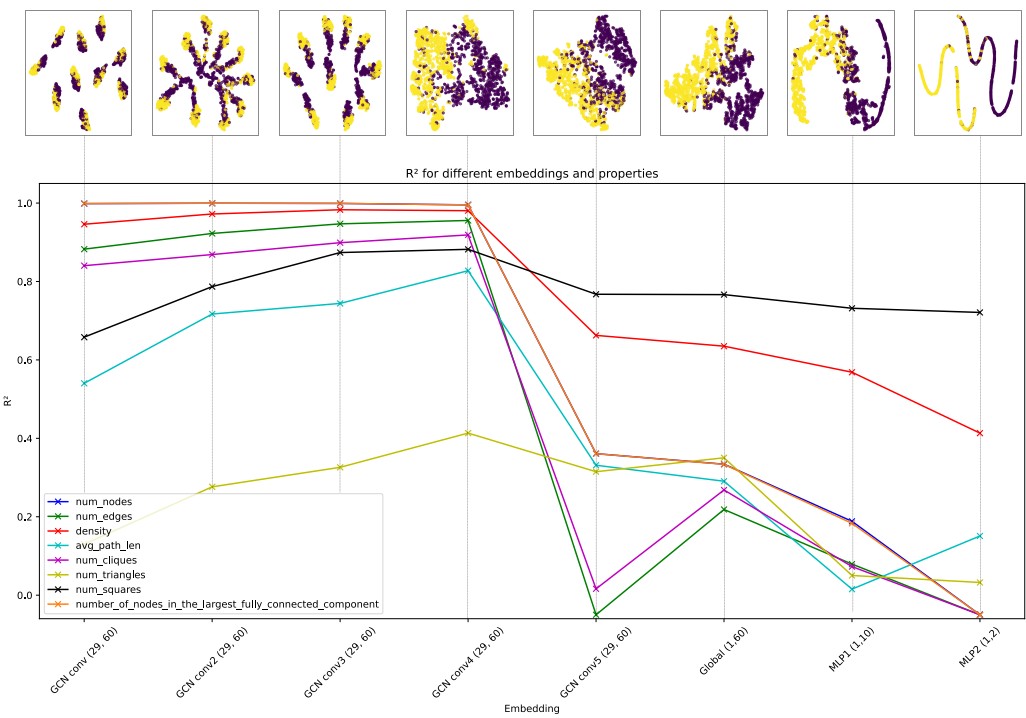

Figure 7: T-SNE visualization across different layers of our GCN architecture aligned with the probing $R^2$ scores plots (Grid House)

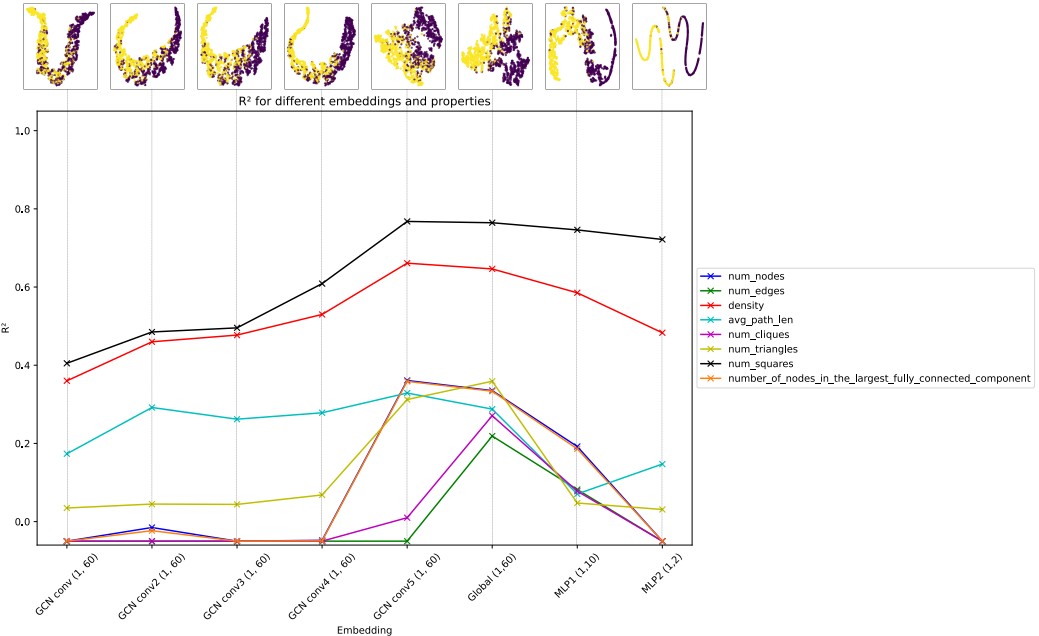

Figure 8: T-SNE visualization across different layers of our GCN architecture aligned with the probing $R^2$ scores plots with mean-pooled node embeddings (Grid House)

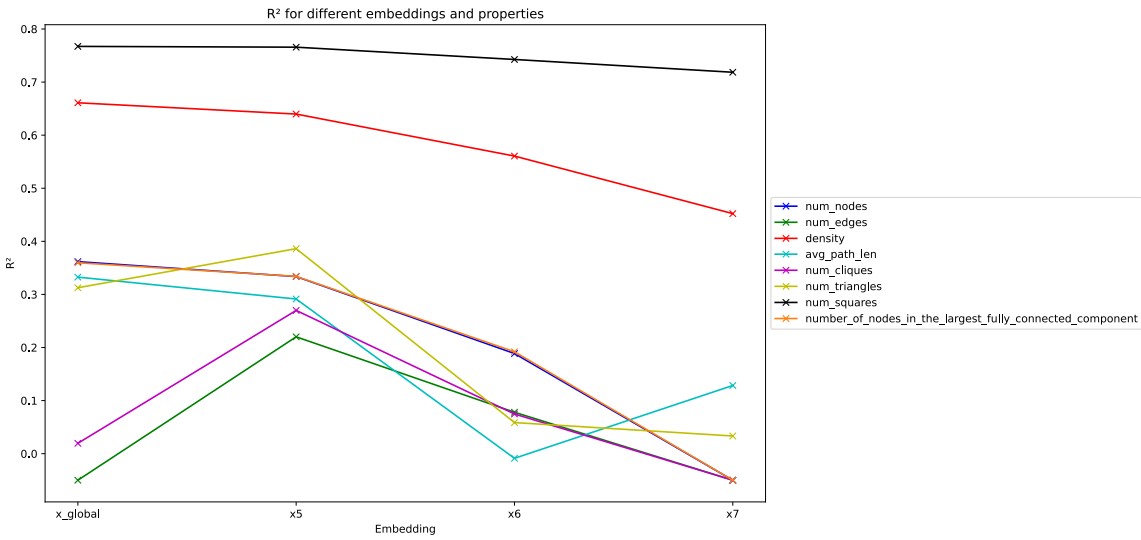

Figure 9: Plot of the GCN (control) $R^2$ results across different layers probing for graph properties with post pooling layers only, allowing clearer visualization and higher order property interpretation (Grid House)

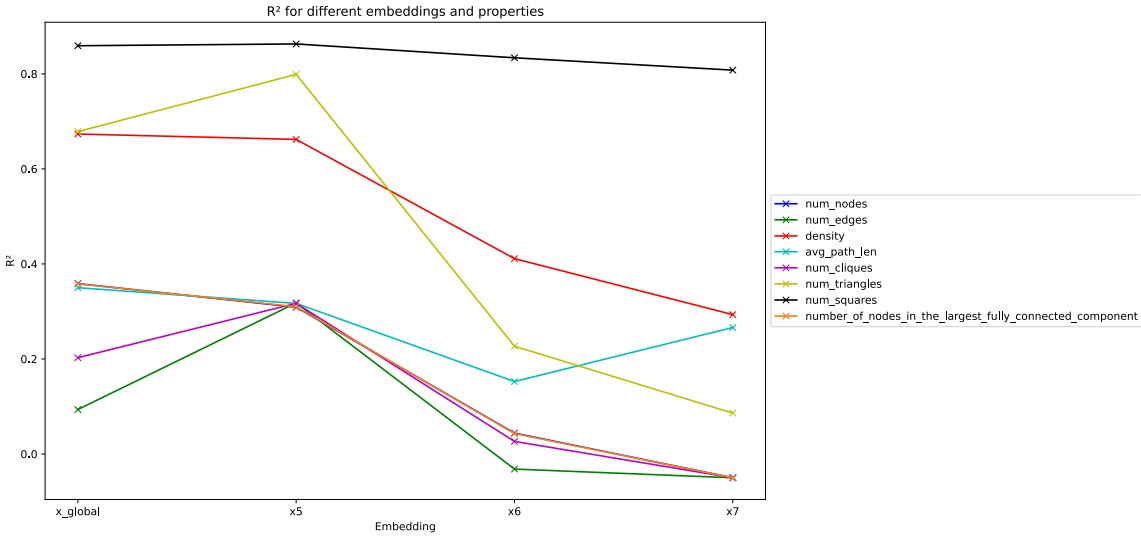

Figure 10: Plot of the GCN ($L_2$) $R^2$ results across different layers probing for graph properties with post pooling layers only, allowing clearer visualization and higher order property interpretation (Grid House)

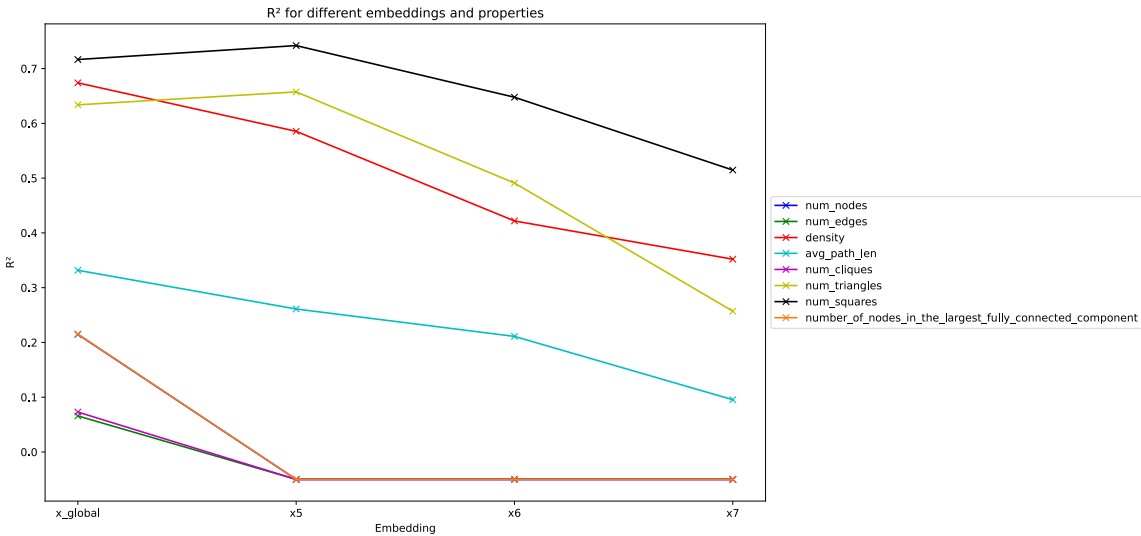

Figure 11: Plot of the GCN (dropout) $R^2$ results across different layers probing for graph properties with post pooling layers only, allowing clearer visualization and higher order property interpretation (Grid House)

**GIN**

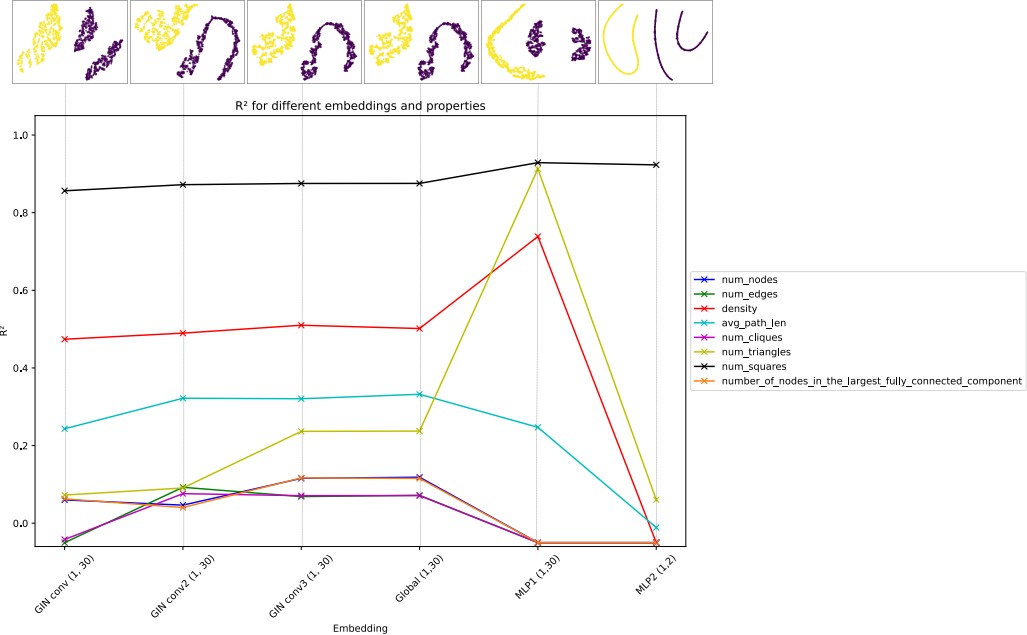

Figure 12: T-SNE visualization across different layers of our GIN architecture aligned with the probing $R^2$ scores plots with mean-pooled node embeddings (Grid House)

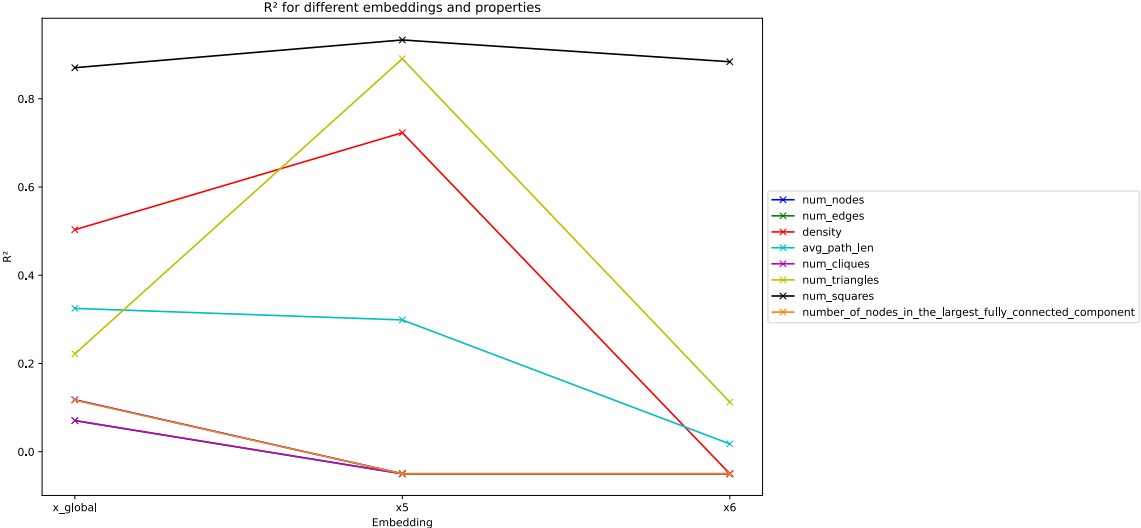

Figure 13: Plot of the GIN (control) $R^2$ results across different layers probing for graph properties with post pooling layers only, allowing clearer visualization and higher order property interpretation (Grid House)

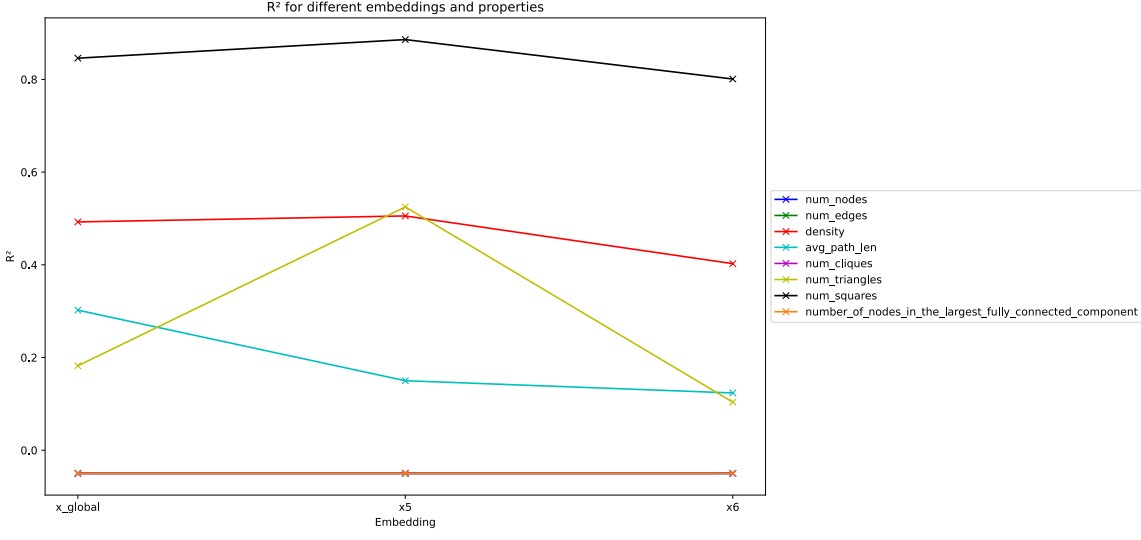

Figure 14: Plot of the GIN ($L_2$) $R^2$ results across different layers probing for graph properties with post pooling layers only, allowing clearer visualization and higher order property interpretation (Grid House)

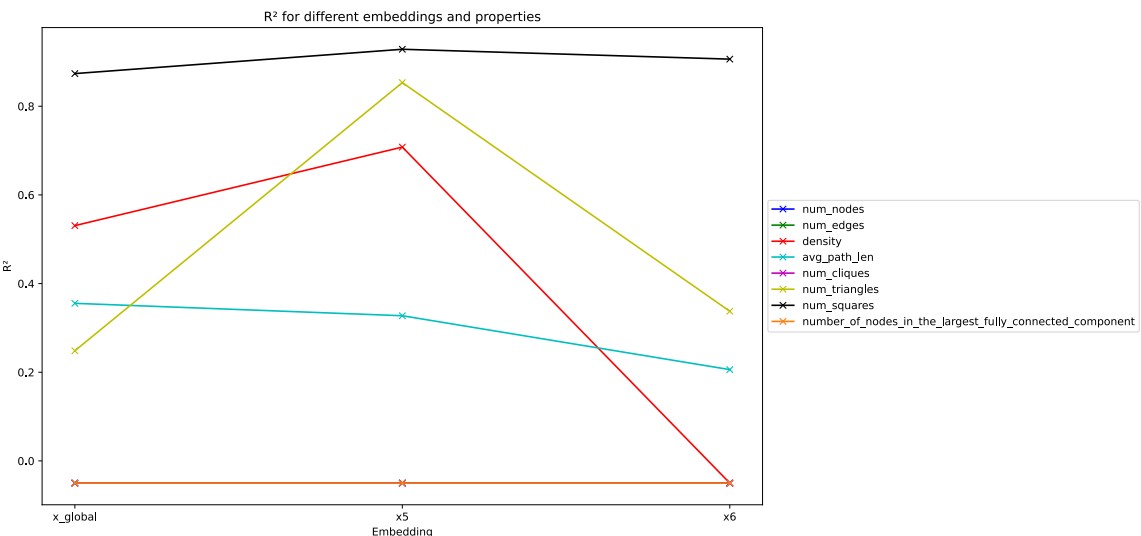

Figure 15: Plot of the GIN (dropout) $R^2$ results across different layers probing for graph properties with post pooling layers only, allowing clearer visualization and higher order property interpretation (Grid House)

**GAT**

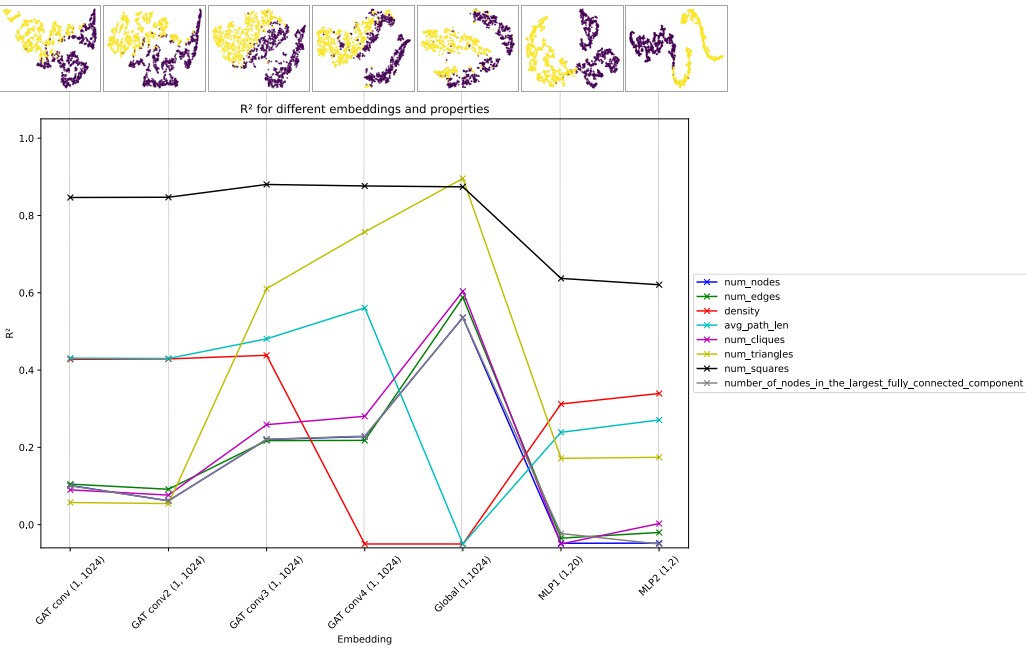

Figure 16:  T-SNE visualization across different layers of our GAT architecture aligned with the probing $R^2$ scores plots with mean-pooled node embeddings (Grid House)

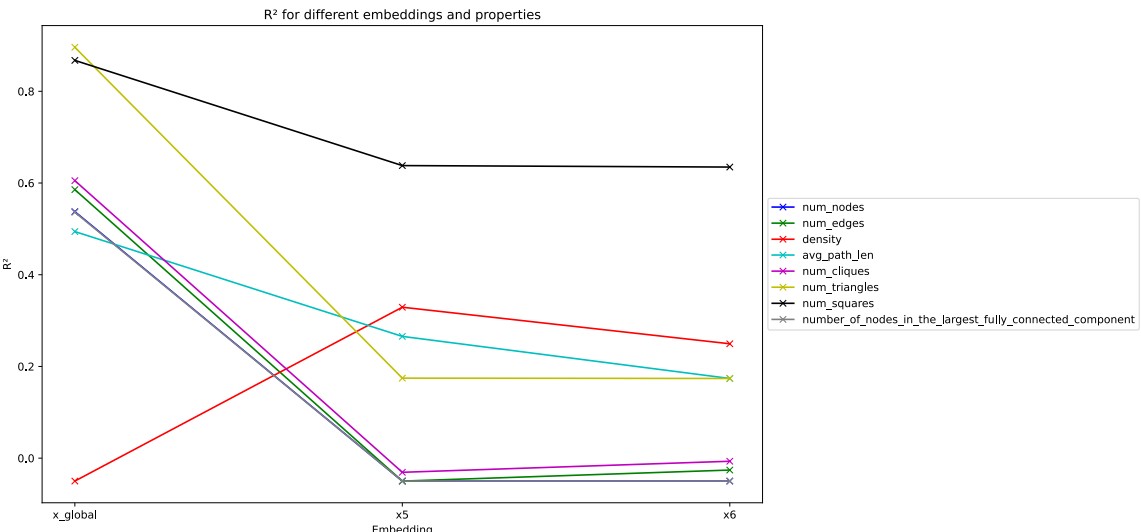

Figure 17: Plot of the GAT $R^2$ results across different layers probing for graph properties with post pooling layers only, allowing clearer visualization and higher order property interpretation (Grid House)

D.3.3. Grid House Node properties probing results

Using the probing method developed in the next section, we were not fully able to confirm our initial hypothesis.

Table 7: Linear Probing $R^2$ Performance Across models for Selected Node Properties (Grid-House Dataset). Best Scores in Bold; Non-convergence indicated by —

| GCN Layer | degree | closeness | betweenness | eigenvector | clustering | pagerank |
|---|---|---|---|---|---|---|
| x1 (GCN) | 0.50 | 0.22 | 0.25 | 0.19 | 0.06 | **0.56** |
| x2 (GCN) | 0.54 | 0.32 | 0.28 | 0.24 | 0.09 | **0.57** |
| x3 (GCN) | 0.54 | 0.35 | 0.29 | 0.25 | 0.11 | **0.57** |
| x4 (GCN) | 0.55 | 0.37 | 0.28 | 0.30 | 0.17 | **0.57** |
| **GIN Layer** | | | | | | |
| x1 (GIN) | 0.55 | 0.18 | 0.24 | 0.22 | 0.05 | **0.56** |
| x2 (GIN) | 0.52 | 0.34 | 0.27 | 0.25 | 0.07 | **0.54** |
| **GAT Layer** | | | | | | |
| Layer 0 | **0.55** | 0.07 | 0.05 | 0.32 | 0.28 | 0.17 |
| Layer 1 | **0.52** | 0.48 | 0.08 | 0.31 | 0.30 | 0.14 |
| Layer 2 | 0.47 | **0.55** | — | 0.29 | 0.29 | — |
| Layer 3 | **0.41** | — | 0.14 | 0.19 | 0.26 | — |
| Layer 4 | 0.35 | **0.50** | 0.12 | 0.21 | 0.23 | — |

In these pre-pooling layers, we first observe the predominance of *page rank* and *node degree* in the early layers and in all the layers of the GCN and the GIN (which has only two of them). When considering the last layers of the GAT (unfortunately we should have have similar architecture with the GIN in order to fully test our hypothesis) it seems that *closeness*, *node degree* and *clustering coefficient* are the most significant. This aligns with our framing of the graph classification task, which is largely driven by the detection of squares and the fact that pre-pooling layers leading to this property detection should affect mostly these three properties. But this does not align with the use of node properties in a graph in order to do graph classification. This still makes a lot of sense. In general, contrary to the graph probing, and to the exception of the node degree, we see that there is not a single property clearly dominating others but that we go towards a combination of different properties just before the graph pooling method. We would have expect the GIN architecture to show similar results with four layers (as we already see an important increase with regard to the closeness between the first and second layer).

# Appendix E. Clintox dataset

## E.1. model

Table 8: Performance of Different Models on ClinTox with a 80%-20% Random Split. The highest performance is highlighted with boldface. All the performance of methods are reported under their best settings.

| Method | ClinTox |
|--------|---------|
| GCN | 0.91 |
| GAT | 0.92 |
| GIN | 0.93 |

## E.2. Results

### E.2.1. GRAPHS PROPERTIES PROBING RESULTS

Table 9: Linear Probing $R^2$ Performance across the GIN layers for basic graph properties (ClinTox dataset). Best Scores in Bold; Non-convergence indicated by —(full)

| GIN Layer | # Nodes | # Edges | Density | Avg. Path Length | Diameter | Radius |
|-----------|---------|---------|---------|------------------|----------|--------|
| x1 (GIN) | **1.00** | **1.00** | 0.66 | 0.76 | 0.55 | 0.60 |
| x2 (GIN) | **1.00** | **1.00** | 0.57 | 0.95 | 0.88** | 0.84 |
| x3 (GIN) | **1.00** | **1.00** | 0.62 | **0.97** | 0.93 | 0.89 |
| x4 (GIN) | **0.99** | **0.99** | 0.37 | 0.91 | 0.82 | 0.82 |
| x5 (GIN) | **0.99** | **0.99** | 0.29 | 0.90 | 0.82 | 0.82 |
| x_global | 0.41 | 0.44 | 0.58 | 0.20 | 0.20 | 0.20 |
| x6 (MLP) | 0.40 | 0.44 | 0.58 | 0.19 | 0.19 | 0.19 |
| x7 (MLP) | 0.42 | 0.46 | 0.50 | 0.27 | 0.23 | 0.25 |
| x8 (MLP) | 0.04 | 0.05 | 0.00 | 0.04 | 0.05 | 0.03 |

Table 10: Linear Probing $R^2$ Performance across the GIN layers for clustering and centrality measures (ClinTox dataset). Best Scores in Bold; Non-convergence indicated by —(full)

| GIN Layer | Clustering coef. | Transitivity | Assortativity | Avg. clustering | Avg. btw. cent. | PageRank cent. |
|---|---|---|---|---|---|---|
| x1 (GIN) | — | — | 0.32 | — | — | 0.18 |
| x2 (GIN) | — | — | 0.21 | — | — | — |
| x3 (GIN) | — | — | — | — | — | — |
| x4 (GIN) | — | — | — | — | — | — |
| x5 (GIN) | — | — | — | — | — | — |
| x_global | — | — | 0.25 | — | 0.48 | **0.40** |
| x6 (MLP) | — | — | **0.27** | — | 0.42 | 0.39 |
| x7 (MLP) | — | — | — | — | **0.47** | — |
| x8 (MLP) | — | — | — | — | 0.06 | — |

Table 11: Linear Probing $R^2$ Performance across the GIN layers for graph substructures (ClinTox dataset). Best Scores in Bold; Non-convergence indicated by —(full)

| GIN Layer | # Cliques | # Triangles | # Squares | Largest comp. size | Avg. degree | Graph energy |
|---|---|---|---|---|---|---|
| x1 (GIN) | 0.99 | — | 0.00 | 0.99 | 0.53 | **1.00** |
| x2 (GIN) | **1.00** | — | 0.00 | 0.99 | 0.46 | **1.00** |
| x3 (GIN) | 1.00 | — | 0.00 | **0.99** | 0.53 | 1.00 |
| x4 (GIN) | 0.99 | — | 0.00 | 0.99 | 0.20 | 0.99 |
| x5 (GIN) | 0.99 | — | 0.00 | 0.99 | — | 0.99 |
| x_global | 0.43 | — | 0.00 | 0.40 | **0.81** | 0.44 |
| x6 (MLP) | 0.43 | — | 0.00 | 0.40 | 0.80 | 0.44 |
| x7 (MLP) | 0.46 | — | 0.00 | 0.42 | 0.75 | 0.46 |
| x8 (MLP) | 0.04 | — | 0.00 | 0.04 | — | 0.05 |

Table 12: Linear Probing $R^2$ Performance across the GIN layers for spectral and small-world properties (ClinTox dataset). Best Scores in Bold; Non-convergence indicated by —(full)

| GIN Layer | Spectral rad. | Algebraic co. | Small world coef. | Small world idx | Avg. btw. cent. |
|---|---|---|---|---|---|
| x1 (GIN) | 0.70 | 0.78 | — | — | — |
| x2 (GIN) | 0.66 | **0.80** | — | — | — |
| x3 (GIN) | 0.61 | 0.80 | — | — | — |
| x4 (GIN) | 0.16 | 0.78 | — | — | — |
| x5 (GIN) | — | 0.69 | — | — | — |
| x_global | **0.74** | 0.67 | — | — | 0.48 |
| x6 (MLP) | 0.74 | 0.66 | — | — | 0.42 |
| x7 (MLP) | 0.71 | 0.56 | — | — | **0.47** |
| x8 (MLP) | 0.07 | 0.02 | — | — | 0.06 |

## E.2.2. Plots

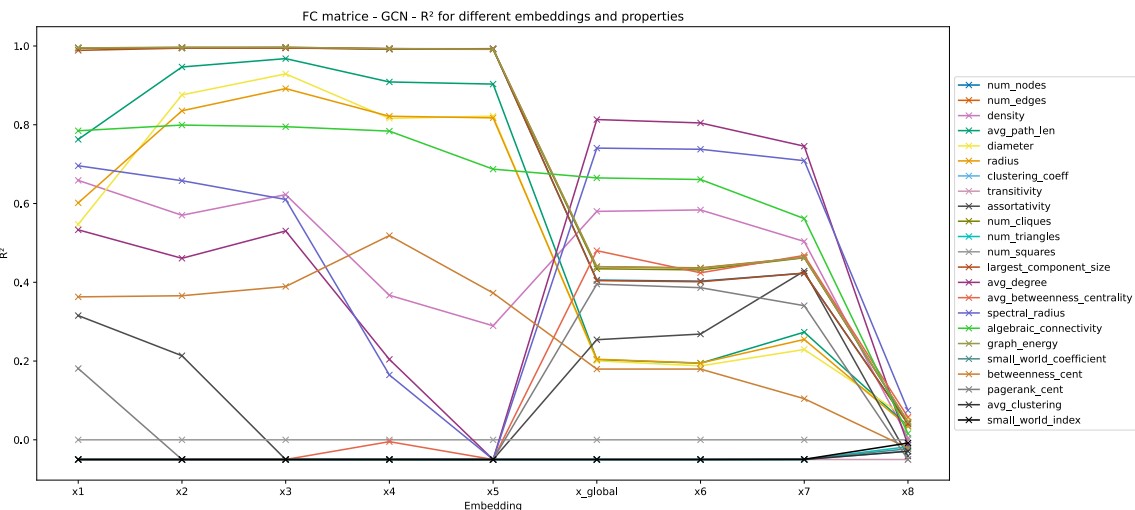

Figure 18: Plot of the GIN $R^2$ results across different layers probing for graph properties. ClinTox dataset (the negative $R^2$ values have been reduced to -0.05).

E.2.3. Node properties probing results

Table 13: Linear Probing $R^2$ Performance across the GIN layers for various node properties (ClinTox dataset). Best Scores in Bold; Non-convergence indicated by —

| GIN Layer | degree | closeness | betweenness | eigenvector | clustering | pagerank |
|---|---|---|---|---|---|---|
| x0 (GIN) | **0.99** | 0.06 | 0.57 | 0.30 | — | 0.16 |
| x1 (GIN) | **0.85** | 0.12 | 0.51 | 0.31 | 0.00 | 0.20 |
| x2 (GIN) | **0.89** | 0.11 | 0.59 | 0.29 | — | 0.26 |
| x3 (GIN) | **0.86** | 0.07 | 0.51 | 0.28 | — | 0.17 |
| x4 (GIN) | **0.85** | 0.09 | 0.49 | 0.32 | — | 0.14 |

Here again, the very strong presence of the node degree makes a lot of sense when we know this property prepares the aggregation of global properties in the post pooling layers. The interesting thing is the non negligible presence of the betweenness centrality in all the layers which suggests that the betweenness centrality of atoms is important in the aggregation of global molecule properties that help predict the toxicity of a molecule. This property is more than the closeness or the clustering coefficient. The irreplaceable nature of some atoms in the molecular graph, which is literally the meaning of having a high betweenness centrality, is an important feature which makes these atoms targets to be part of higher order molecular schemes and patterns.

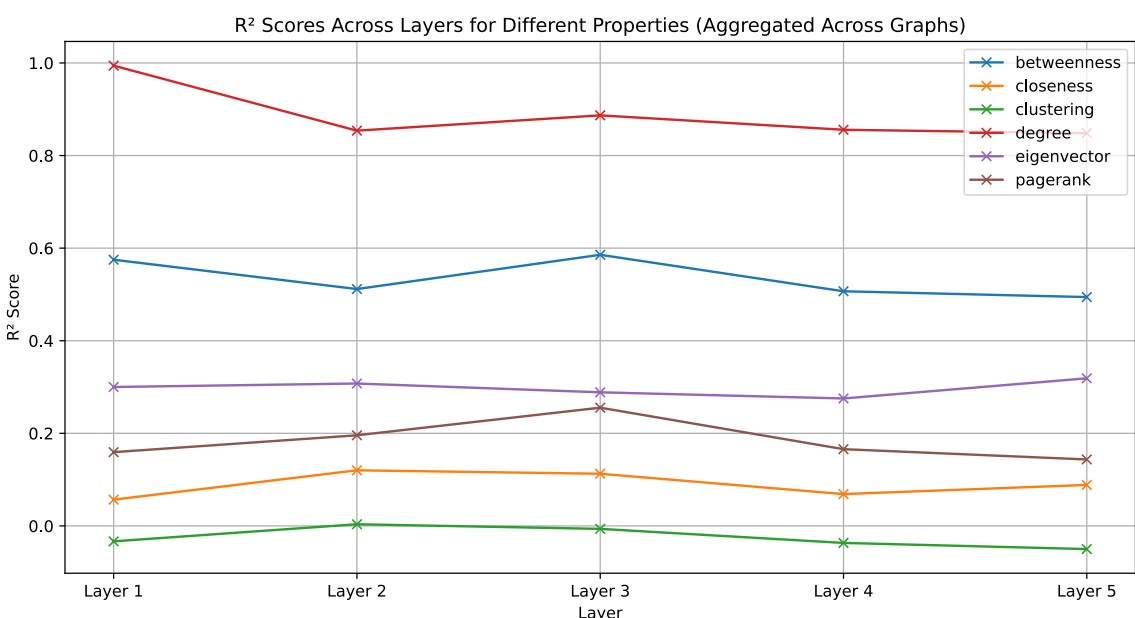

Figure 19: Plot of the GIN $R^2$ results across different layers probing for node properties. ClinTox dataset (the negative $R^2$ values have been reduced to -0.05). (full results)

