# OpenReview forum: "Do Graph Neural Network States Contain Graph Properties?"
_nesyconf.org/NeSy/2025/Conference_Phase_2 — NeSy 2025 - Phase 2 Poster_

### Official Review · Reviewer_sMeh · 2025-07-02
**They might, specially if the data encoded contains graph properties/relations**

**Rating:** 6
**Confidence:** 2

**Review:**

This paper presents a model agnostic explainability pipeline for Graph Neural Networks (GNNs) employing diagnostic classifiers. It considers graph-theoretic properties as the features of choice for studying
the emergence of representations in GNNs. This pipeline aims to interpret the learnt representations in GNNs.

Different GNN architectures are tested regarding their ability to solve a graph classification problem through optimal feature extraction. They render it linearly separable in the space of their embeddings through the computation of graph properties.

This paper validates domain knowledge with the Clintox
Molecular dataset. There is a manifest emergence of molecular qualities like toxicity with respect to their structural properties like node degree (atom valency) and spectral radius (the molecule’s stability).

This paper is 59 pages long. The authors must do an effort to summarize the main content.

**Anonymity:**

Remain anonymous

---

### Official Review · Reviewer_YJuC · 2025-07-07
**This study examines GNN embeddings to determine whether training captures structural graph characteristics as degree centrality, path length, and cycle count. The separation of graph-theoretic features in latent representations is evaluated using diagnostic linear classification methods in this model-agnostic probing method. Two datasets: the ClinTox molecular dataset and a synthetic Grid-House dataset intended to evaluate structural reasoning are used for the experiments. The findings are consistent with the theory that GNNs gradually integrate symbolic graph features in an interpretable and layer-wise manner. Given the lack of a code repository in the manuscript, along with some lack of details regarding hyperparameter selection, I grade the paper with a 6:: Marginally above acceptance threshold.**

**Rating:** 6
**Confidence:** 4

**Review:**

### Quality:

* The inclusion of two datasets, one synthetic and one real-world, helps contextualize the results, and the experimental design is logically solid.   Nevertheless, replication is presently not possible without providing a code repository, and several crucial elements (such as probe configuration and model selection mechanism) are not sufficiently explained. How did the GNN models' and the probing classifiers' hyperparameters were chosen? Was this done via grid search, prior defaults, or manual tuning? If manual, what criteria were used?


### Clarity:

* Diagrams and tables are easy to follow, background information is adequate.

* However, the  hyperparameter selection process is an important aspect that is lacking detail. While they are providing those numbers, the whole decision as to why they were selected is not in the manuscript.

* Related work as a separate section would be gladly appreciated instead of being in the Appendix in order to fully demonstrate the strengths of the chosen methodology over past ones and an overall motivation for the work in order to ease the readability and not go back and forth.

### Originality:

* The adaptation of the NLP probing framework (e.g., internal activation linear classifiers) to the GNN context is a fresh contribution. Although this paper draws inspiration from previous research on probing for graph motifs, it expands the concept in a more methodical manner across a variety of characteristics, layers, and models.


### Significance:

* For both theoretical understanding and model debugging, it offers tangible evidence that important structural characteristics are embedded in hidden states.

* However, without a code repository it is hard to fully go through the proposed methodolgy, pipelines and results hence there is some weakness in the "soundness" of the paper.


### Pros:

* The probing results, especially on the Grid-House dataset, show a strong correlation between GNN embeddings and interpretable features such as the number of cycles or average degree.

* The probing method is proposed to be applied to a wide variety of GNN architectures and datasets, making it flexible.

* The authors are able to confirm if models identify motifs algorithmically by using the carefully designed Grid-House dataset to evaluate structural consistency.

### Cons:

* The background and related work part discusses the fundamentals of GNN and XAI, however it doesn't go into great detail about other tools like GraphMask, SubgraphX, PGExplainer, or GNNExplainer. What are the scope and capabilities of your approach that set it apart from these current GNN justification techniques? How does your work clearly differentiate and how is your method leveraged?

* Why did you choose to report only the best run? Can you provide average and variance metrics across all runs to better represent performance stability?

* Replication and clear documentation is key when it comes to adopting different methodologies. Why was not a code repository linked in the manuscript? Technical soundness cannot be fully evaluated hence some reliability is at loss.

**Anonymity:**

Remain anonymous

---

### Official Review · Reviewer_2CpB · 2025-07-09
**solid empirical study about GNN representations**

**Rating:** 7
**Confidence:** 3

**Review:**

The paper analyzes to what extent standard GNN architectures (GINs and GATs)
encode global graph properties in their learned representations using two
datasets, one synthetic and one real-world.  The results indicate that
task-relevant properties can be linearly decodable from the top layer and that
other properties are linearly decodable from the lower layers.


#Clarity

- Generally well written and easy to follow.  See below for typos (I didn't
  find many).


#Significance

- I believe this work could be of interest to GNN practitioners; I know some
  of them are very much interested in how GNNs handle/encode global graph properties.

- I am not entirely sure this paper touches on "NeSy approaches for GNNs", as
  listed on the NeSy website.  That is, it may not be entirely appropriate for
  the conference.


#Originality

- The work appears to be novel (but I am no expert).

- Well positioned against relevant literature.


#Quality

- This is primairly an empirical investigation.  The experimental setup
  is described in enough detail and it appears to be sound.

- The discussion of results seems to be accurate.

- "We manifest both the expressivity of different GNN architectures and their
  ability to solve a graph classification problem through optimal feature
  extraction" - I can agree with expressivity, but what's optimal in their
  behavior? Is this supported by evidence? The best decodability result sits
  around 80% R2.  This doesn't feel optimal to me.  Please rephrase.


#Remarks

- It's good to keep in mind that the "linear representation hypothesis" is just
  a hypothesis, with evidence both for and against (see the LLM literature).
  Actually, I think it's the other way around: it's not that linear probes are
  justified by the LRH, it's that they can be used to *check* if the LRA
  holds in the particular model/dataset.

- "We propose a pipeline to confirm that graph-theoretic properties are a good
  candidate for studying GNN inductive bias": is this what the paper is about?

- "reordering the nodes does not affect their norms" - this depends on the norm
  you choose, think Mahalanobis.

- There are two interesting (to me) findings that are not highlighted in the
  main text:

  - some global properties are linearly decodable from the bottom layers. This
    is surprising because they appear not to be task relevant, so it is not
    clear to me why the network should learn them - they are essentially
    discarded by the higher layers. One option is they somehow contribute to
    predicting task-relevant properties that survive till the top layer, but I
    don't think this is the case, because the task-relevant properties are
    decodable with high accuracy even at the lower layers. Another option is
    that the particular way in which features are fed to the linear probe makes
    it possible to predict these properties linearly, but I'm not sure this
    makes sense.

  - going up the layer ladder, some undecodable properties become suddenly
    decodable.  This suggests the GNN "unfolds" or "linearizes" representations
    at the top layers that were previously mixed by the underlying layers. It's
    unclear why the network should do that - it feels suboptimal.

  Neither observation compromises the message of the paper.

- A few typos. E.g.,

 - "make them as expressive while We".

 - "Important note :" - extra space.

 - "some few properties"

**Anonymity:**

Remain anonymous